# Factors influencing use of essential surgical services in North-East India: a cross-sectional study of obstetric and gynaecological surgery

Tim Ensor [iD],[1] Amrit Virk [iD],[1] Noel Aruparayil [iD] [2]

[1]Nuffield Centre for International Health and Development, University of Leeds, Leeds, UK
[2]Leeds Institute of Medical Research at St James', University of Leeds, Leeds, UK

**Correspondence to**
Professor Tim Ensor;
T.R.A.Ensor@leeds.ac.uk

## ABSTRACT

**Introduction** There continues to be a large gap between need and actual use of surgery in low-resource settings. While policy frequently focuses on expanding the supply of services, demand-side factors are at least as important in determining under utilisation and over utilisation. The aim of this study is to understand how these factors influence the use of selected essential obstetric and gynaecological surgical procedures in the underserved and remote setting of North-East India.

**Methods** The study combines and makes use of data from a variety of surveys and routine systems. Descriptive analysis of variations in caesarean section, hysterectomy and sterilisation and then multivariate logit analysis of demand-side and supply-side factors on access to these services is undertaken.

**Results** Surgical rates vary substantially both across and within North-East India, correlated with service capacity and socioeconomic status. Travel times to surgical facilities are associated with rates of caesarean section and hysterectomy but not sterilisation where services are much more deconcentrated. Travel is less important for surgery in private facilities where capacity is much more dispersed but dominated by the non-poor. The presence of non-doctor medical staff is associated with lower levels of surgical activity.

**Conclusion** In low resource, remote settings policy interventions to improve access to services must recognise that surgical rates in low-resource settings are heavily influenced by demand-side factors. As well as boosting services, mechanisms need to mitigate demand-side barriers particularly distance and influence practice to encourage surgical intervention only where clinically indicated.

## Strengths and limitations of this study

► Combines three large data sets on obstetric services to allow an exploration of the supply and demand factors affecting use of obstetric and gynaecological services.
► Makes use of a distance variable that allows consideration of the effect of location on demand for services.
► The distance variable is a rough estimate of the actual time to a facility because of the geographical displacement in the Geographic Iniformation System variable and the assumption that facilities are based in the centre of the district in the absence of precise GIS coordinates for facilities.
► Demographic and Health Survey data do not indicate whether surgery was clinically indicated at an individual level, so drawing definitive conclusions on excessive surgery at an individual level is therefore not possible.
► Relies on self-reported use of surgical services.

global burden of disease.[4] Improved surgery is a key component in strengthening a system to deliver universal health coverage.[5 6]

Much recent focus on access to global surgery has been on strengthening the supply side through increasing surgical staff, improving training and infrastructure provision.[7 8] A strong supply side is necessary but it is not sufficient for improved access. A host of demand-side factors at the user level such as affordability, proximity to services, socioeconomics and education influence service use.[9] Understanding how these factors influence surgical access is crucial for the design of evidence-based policy interventions, yet there is limited evidence on the size of their impact particularly in low-resource settings.[2 10 11]

This study aims to examine the extent to which demand and supply factors influence surgical use in such underserved, poorly resourced and remote settings. Our focus is on the eight states in North-East India, one

## INTRODUCTION

There continues to be a huge difference between user need and use of surgical care in low-resource settings with up to 4.8 billion lacking accessible services.[1 2] Use of services is typified by substantial inequality with both underuse and overuse of surgical services harming human health.[3] It is estimated that increased appropriate access to surgical services could have a substantial impact on the

of the most underdeveloped regions in the country with persistently poor health outcomes and substantial access barriers.[12] We examine factors determining use of three essential surgical procedures: caesarean sections (CSs), hysterectomy and female sterilisation.[13] Although some of these factors are likely to be specific to this type of surgery, others can be regarded as more general. In this respect, these procedures act as tracer services to identify general issues of access to surgical provision.

## METHODS

### Data sources

This cross-sectional study based on secondary data sources makes use of three data sets. The National Family Health Survey (NFHS) 2015–2016 provides a sample of 98 700 women of reproductive age that is statistically representative down to district level.[14] The online Health Management Information System (HMIS) provides information on surgical activity and was used to identify centres with surgical capability.[15] The District Census provides supply-side information on numbers of facilities, staff, beds and population by district.[16]

### Variables and method of analysis

Descriptive statistics on state and district level surgical activity in North-East India are compared with other states and presented to set the regional discussion in context. Subsequently, for the eight northeastern states (Arunachal Pradesh, Assam, Manipur, Meghalaya, Mizoram, Nagaland, Sikkim and Tripura) multivariate logit analysis is used to examine the association between use by women of three surgical procedures, individual and household characteristics (age, education, parity, caste/tribe, marital status, insurance held and economic status), proximity to and capacity of health services. The p values for the key model coefficients were estimated based on the t-statistic with 95% significance used as the threshold for valid inference. The stata margins command is used to estimate the relationship between the outcome and individual factors holding other independent variables at their mean values.[17] The economic status of the household is approximated by the asset-based wealth index.[18] Three supply variables are used, available at the village level from the district-level census: doctors, other medical staff and beds expressed as rate per 1000 population.

The distance between clusters of households and main surgical centres is computed through a two-stage process. The NFHS provides approximate geographical positioning of the cluster from which households are selected. In line with the procedure adopted across Demographic and health Survey (DHS), coordinates are then shifted randomly by up to 5 km in rural and 2 km in urban areas to protect the anonymity of responding households. HMIS data provide information on the amount of surgical activity in facilities within a district allowing computation of the size of their surgical capacity defined as high (more than 2000 major surgical procedures a year) through to

low (less than one per day). The time it takes to travel from the centre of each district to all other districts with high surgical activity is calculated in Stata using the georoutei add-in.[19] This takes pairs of coordinates (longitude and latitude) and accesses the HERE mapping database (developer.here.com) to compute the total travel time using available roads adjusted for speed of traffic. We calculated the distance between each district centre and the nearest district with surgical capacity which was then added to the distance between each household cluster and district centre, producing an estimate of total time to a surgical centre.

## RESULTS

Nationally, the HMIS reports a total of 4.3 million (3.1 per 1000 population) major (with spinal/general anaesthesia) and 11.5 million (8.5 per 1000 population) minor surgical procedures in 2017–2018. The rates vary from 1.2/1000 major surgical procedures in the country's most populous state Uttar Pradesh to 18/1000 in largely urban city-state Delhi. Rates are much lower than the needs based benchmark of 45/1000 major surgical procedures for South Asia.[1] With the exception of Mizoram state, where health service infrastructure appears to be better functioning than in other states,[20] rates of surgery across the North-East are at or below the country average of 3.1 per 1000 population (figure 1).

The national CS rate is reported by place of service focused HMIS as 17.5% of live births for 2017–2018 while the household focused NFHS reports a similar figure.[14] This average, which is higher than the WHO benchmark for necessary surgical delivery of 10%–15%,[21] masks substantial state-wide variation ranging from 2.5% in Bihar to over 45% in Tamil Nadu.[15] The NFHS reported 31% of Indian women that had a sterilisation which is close to the highest rate internationally—33% in the Dominican Republic.[22] Comparing hysterectomy rates across countries is impeded by the nature of the survey data. The NFHS focuses on women of reproductive age whereas in established health systems hysterectomy is largely concentrated in older age groups. Studies in India have drawn attention to the relatively high rates of the procedure with one-third occurring in women below the age of 40.[23] In NE India, rates of gynaecological and obstetric surgery are at or below the national average: CS rates range from 7.9% (Meghalaya) to 21.8% (Tripura) of deliveries (national average=17.5%), hysterectomy from 0.2 (Assam) to 2.8 (Arunachal Pradesh) per 1000 women (national average=0.8) and sterilisation from 0.2 (Manipur) to 1 (Assam) per 1000 women (national average=2.6).

Turning to surgical utilisation, across India use of surgery (per 1000 population) is positively associated both with the capacity of health services and the wealth of individuals (table 1) with associations strongest for CS. The stronger association with the total number of registered doctors (per 1000 population) in the state rather than

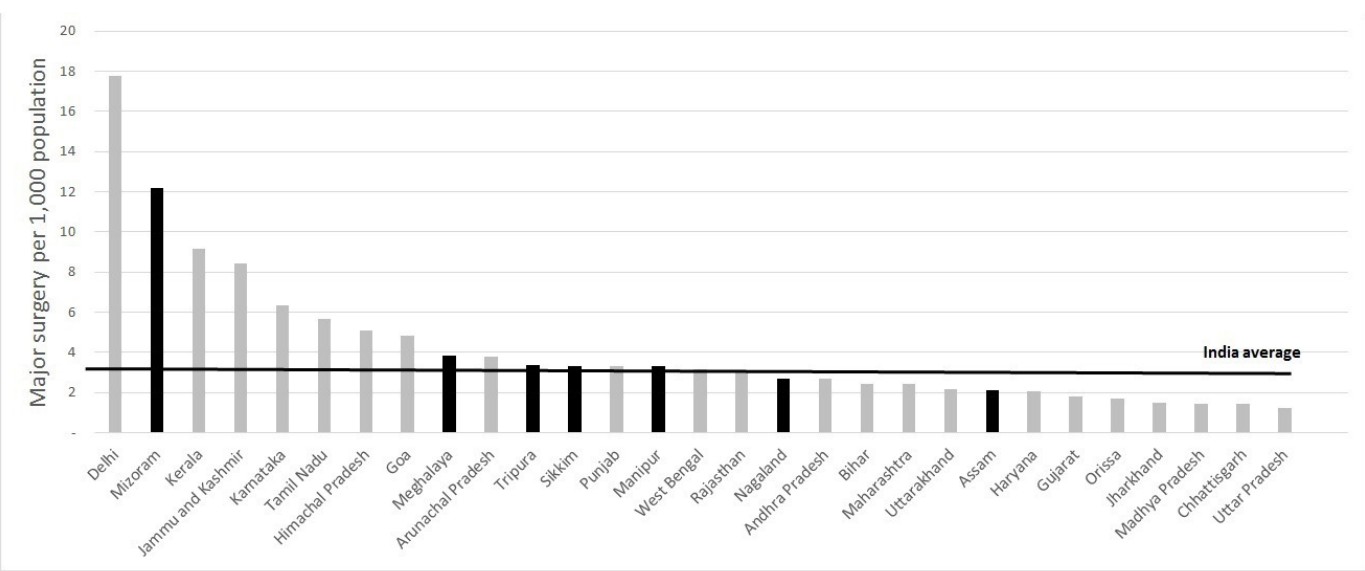

**Figure 1** Major surgeries per 1000 population across states of India (2016–2017), bars shaded black are in North East India.

number of government doctors (per 1000 population) may reflect the importance of the private sector in delivering the majority of CS across the country. According to the HMIS, 51% of CS were delivered in private facilities in 2016–2017, while the NFHS reports 53% for births in 2014–2016. The association with health service capacity and individual wealth is much weaker for hysterectomy and sterilisations. Sterilisation (per 1000 women) is positively associated with the total number of government doctors but not total registered doctors in states, indicating public sector dominance over the provision of permanent contraception methods with 74% provided in the public sector.[15]

Across NE India, use of surgical services varies widely: CS as a proportion of deliveries by district varies from 0.4% to almost 42% (IQR 8.7%), hysterectomy from 0.1% to 4.8% (IQR 1.2%). Rates of female sterilisation vary from 0.5% to more than 21% (IQR 5.7%) (figure 2).

Surgical use is positively associated with proximity to major surgical centres. CS rates, for example, range from over 25% for clusters than are geographically close to major surgical centres to 5% or lower in areas more than 10 hours away (figure 3). There is a less pronounced relationship between distance and prevalence of hysterectomy (figure 4), while no clear association is found for sterilisation.

Turning to the analysis of individual behaviour, all three surgical procedures are positively associated with household wealth (table 2). This association is strongest for CS and weakest for sterilisation. CS rates are substantially higher for women who are better educated. In contrast, hysterectomy and sterilisation rates are lower for those with secondary education or above compared with those without any education.

The importance of economic status is likely to be partly the result of cost barriers to obtaining care. Costs of

| Table 1 | Correlation between surgery, income per capita and supply-side variables across all states of India | | | |
|---|---|---|---|---|
| | Surgical procedures per 1000 population | C-section rate (% live births) | Hysterectomy per 1000 women | Sterilisations per 1000 women |
| All India | 3.1 | 17% | 0.8 | 2.5 |
| NE India states | 2.8 | 18% | 0.5 | 0.8 |
| $R^2$ statistics | | | | |
| Registered doctors per 1000 | 0.373* | 0.612* | 0.179† | 0.003 |
| Government doctors per 1000 | 0.281† | 0.234† | 0.102‡ | 0.252† |
| Beds per 1000 | 0.288† | 0.336† | 0.090‡ | 0.094‡ |
| Gross Domestic Product per capita | 0.103‡ | 0.280† | 0.047 | 0.020 |

*Significant at the 1% level.
†Significant at the 5% level.
‡Significant at the 10% level.

**C-section at last birth**

Q1    Q3

0.0%    5.0%    10.0%    15.0%    20.0%    25.0%    30.0%    35.0%    40.0%    45.0%

**Have had a sterilisation**

0.0%    5.0%    10.0%    15.0%    20.0%    25.0%

**Have you had your uterus removed**

0.0%    1.0%    2.0%    3.0%    4.0%    5.0%    6.0%

Percentage of women reporting

**Figure 2**  Proportion of women that reported undergoing three gynaecological/obstetric procedures.

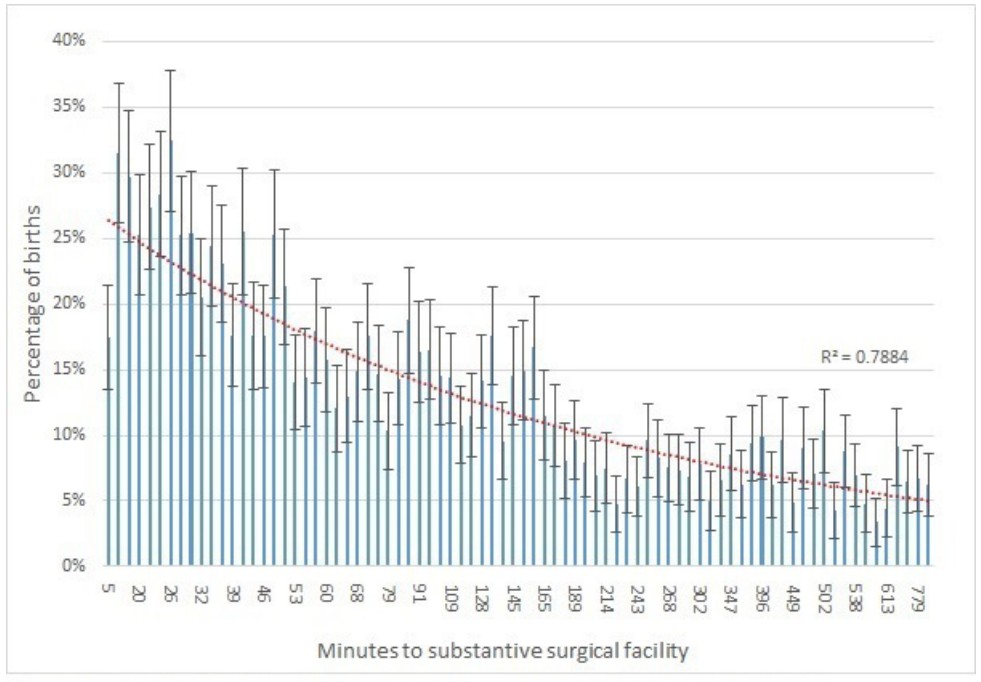

**Figure 3**  Variation in CS by distance from substantial surgical facility. CS, caesarean section (error bars are 95% confidence intervals).

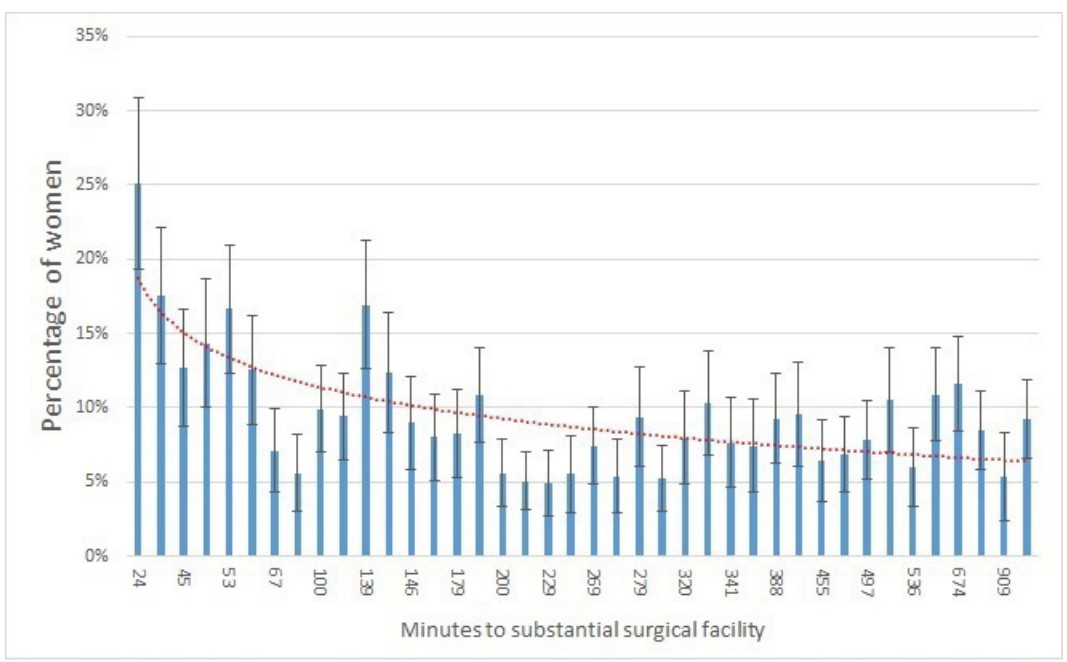

**Figure 4** Variation in hysterectomy (prevalence) by distance from substantial surgical facility. (error bars are 95% confidence intervals)

surgical delivery, including transport represent a substantial proportion of per capita income; 11% in the public sector and 28% in the private sector (table 3). The NFHS found that after distance (36%), treatment cost was the second most important factor (21%) for women not delivering in a facility. In contrast, sterilisation costs were reportedly less burdensome for users as 75% of sterilised women in the NFHS reported not paying anything and 48% reported receiving financial compensation.[14]

The incidence of CS increases with age particularly for those over 30 compared with younger women. The prevalence of hysterectomy and sterilisation is also positively associated with age the latter likely reflecting an increased demand for birth control as family size grows. Higher parity is also associated with greater prevalence of sterilisation. The analysis suggests that women with Muslim or Christian faith are less likely to undergo the three procedures while those living in tribal communities have lower levels of CS than other communities. There is no association with the other gynaecological surgery.

RSBY (Rashtriya Swasthya Bima Yojana) insurance is the most common insurance in the NE states with the exception of Arunachal Pradesh and Sikkim where state insurance dominates. RSBY membership is negatively associated with CS and has no statistically significant association with hysterectomy. Sterilisation is positively associated with RSBY as well as other types of insurance (table 2).

Distance to a major surgical centre is negatively associated with both CS and hysterectomy but no statistically significant relationship is found for sterilisation. No information on where sterilisation was carried out is available in the NFHS and those reporting place of hysterectomy is too small for a meaningful analysis. The CS rates vary

substantially across the public and private sectors for all income groups. Overall CS rates exceed 40% for women delivering in private facilities falling to 11% in public facilities. There is a strongly pronounced income gradient for both facility types (figure 5).

Proximity to medical services, as measured by doctor numbers, is shown to be positively associated with CS and hysterectomy rates. Availability of more local doctors may also increase referral to facilities even if not situated in the area. Local bed capacity is associated with CS rates but not with hysterectomy. In fact, across the eight states, provision of hysterectomy is highly concentrated with just 12% of districts carrying out almost 70% of hysterectomies in 2017/2019 (HMIS).

The association with distance from a main surgical facility is notably different in public and private facilities. For public facilities, there is a large and statistically significant difference between women living close versus those living far from a large surgical facility (figure 5). In contrast, for women delivering in private facilities, the effect of proximity is not nearly so pronounced and not significant for any income group.

## DISCUSSION
### Key findings
Surgical rates vary substantially both across and within the states of North-East India, correlated with service capacity and socioeconomic status. The results demonstrate the importance of individual level demand-side factors on the use of surgical services across NE India. The strong positive relationship between economic and educational status and CS is evidenced in other studies in South Asia and across Low Middle Income Countries.[24 25] Like recent

**Table 2** Association between use of surgical services and individual, household and supply-side characteristics (multivariate logit with cluster adjusted standard errors)

| | Caesarean section | | | Hysterectomy | | | Sterilisation | | |
|---|---|---|---|---|---|---|---|---|---|
| | Coef. | t-statistic | Sig. | Coef. | t-statistic | Sig. | Coef. | t-statistic | Sig. |
| Wealth group (base=poorest) | | | | | | | | | |
| Poorer | 0.483 | 3.840 | * | 0.495 | 2.480 | † | 0.013 | 0.190 | |
| Middle | 1.093 | 8.390 | * | 0.739 | 3.610 | * | 0.105 | 1.380 | |
| Richer | 1.629 | 11.660 | * | 1.162 | 5.480 | * | 0.184 | 2.010 | † |
| Richest | 2.072 | 13.340 | * | 1.330 | 5.480 | * | 0.459 | 3.740 | * |
| Education (base=no education) | | | | | | | | | |
| Completed primary | 0.552 | 3.660 | * | 0.301 | 1.900 | ‡ | −0.012 | −0.170 | |
| Completed secondary | 0.834 | 6.370 | * | −0.080 | −0.590 | | −0.382 | −5.460 | * |
| Completed higher | 1.230 | 7.950 | * | −0.253 | −0.980 | | −0.931 | −7.100 | * |
| Parity (base=one delivery) | | | | | | | | | |
| Two | −0.641 | −9.060 | * | 0.136 | 0.780 | | 2.432 | 18.630 | * |
| Three | −1.291 | −12.190 | * | 0.162 | 0.880 | | 2.914 | 21.090 | * |
| Four or more | −1.882 | −13.170 | * | −0.145 | −0.780 | | 2.823 | 19.450 | * |
| Marital status (base=never married) | | | | | | | | | |
| Married | −0.763 | −1.380 | | 1.074 | 2.400 | † | 1.078 | 1.350 | |
| Widowed | −0.875 | −1.360 | | 0.579 | 1.180 | | 0.424 | 0.530 | |
| Divorced | −0.694 | −1.050 | | 1.233 | 2.140 | † | 0.473 | 0.560 | |
| Living together/separated | −0.999 | −1.530 | | 0.108 | 0.170 | | 0.619 | 0.730 | |
| Religion (base=hindu) | | | | | | | | | |
| Muslim | −0.660 | −5.840 | * | −0.723 | −3.230 | * | −1.649 | −14.340 | * |
| Christian | −0.281 | −2.510 | † | −0.014 | −0.070 | | −0.298 | −2.850 | * |
| Sikh | 3.705 | 3.090 | * | – | – | | 0.273 | 0.200 | |
| Buddhist/neo-buddhist | −0.671 | −3.850 | * | 0.347 | 0.850 | | −0.225 | −1.350 | |
| Jain | −1.851 | −1.320 | | 3.264 | 2.670 | * | 1.336 | 2.190 | † |
| No religion | 0.436 | 0.740 | | −0.086 | −0.110 | | −1.007 | −2.540 | † |
| Other | −0.111 | −0.860 | | 0.230 | 1.020 | | −1.077 | −9.420 | * |
| Caste/tribe (base=none) | | | | | | | | | |
| Caste | −0.151 | −1.360 | | −0.100 | −0.530 | | 0.165 | 1.620 | |
| Tribe | −0.562 | −4.100 | * | −0.259 | −0.970 | | −0.148 | −1.150 | |
| Age group (base=15–19) | | | | | | | | | |
| 20–25 | 0.589 | 3.460 | * | −2.426 | −3.740 | * | 0.869 | 1.190 | |
| 25–30 | 0.913 | 5.260 | * | −1.229 | −4.290 | * | 1.289 | 1.800 | ‡ |
| 30–35 | 1.311 | 7.190 | * | 0.103 | 0.510 | | 1.671 | 2.320 | † |
| 35–40 | 1.589 | 8.340 | * | 0.821 | 5.430 | * | 1.810 | 2.500 | † |
| 40–45 | 1.613 | 6.630 | * | 0.529 | 4.360 | * | 1.715 | 2.380 | † |
| 45–50 | 1.371 | 2.550 | † | – | – | | 1.607 | 2.220 | † |
| Type of insurance (base=no insurance) | | | | | | | | | |
| State | 0.072 | 0.580 | | 0.017 | 0.120 | | 0.226 | 2.550 | † |
| RSBY | −0.254 | −2.170 | † | 0.092 | 0.470 | | 0.266 | 3.550 | * |
| Other insurance | 0.252 | 1.100 | | 0.008 | 0.020 | | 0.434 | 2.800 | * |
| Supply-side variables | | | | | | | | | |

Continued

**Table 2** Continued

| | Caesarean section | | | Hysterectomy | | | Sterilisation | | |
|---|---|---|---|---|---|---|---|---|---|
| | Coef. | t-statistic | Sig. | Coef. | t-statistic | Sig. | Coef. | t-statistic | Sig. |
| Doctors/1000 population | 0.022 | 1.670 | ‡ | −0.043 | −1.670 | ‡ | 0.031 | 3.090 | * |
| Other medical/1000 population | −0.021 | −2.930 | * | 0.011 | 0.990 | | −0.044 | −7.050 | * |
| Time to main surgical facility | −0.001 | −3.550 | * | −0.001 | −2.340 | † | −0.000 | −0.710 | |
| Beds/1000 population | 0.017 | 3.690 | * | 0.005 | 0.720 | | 0.025 | 6.620 | * |
| Constant | −2.494 | −4.120 | * | – | – | | −6.695 | −6.320 | * |

*Significant at the 1% level.
†Significant at the 5% level.
‡Significant at the 10% level.
RSBY, Rashtriya Swasthya Bima Yojana.

studies on hysterectomy in India,[23] we find a strong association between hysterectomy rates and household wealth. Higher education, however, is less influential, a finding borne out by another study in India reporting higher hysterectomy rates among uneducated, rural women.[11] Turning to sterilisation, economic status is not a strong predictor for prevalence but our results indicate that those with at least secondary education are less likely to be sterilised. Within groups, even after adjusting for distance and socioeconomic factors, CS rates are still higher among non-tribal compared with tribal women, a finding supported by a recent study in western India.[26] Capacity of facilities is associated with higher CS rates but not hysterectomy. Hysterectomy is typically a more complex procedure requiring more resources, so it is unsurprising that increased capacity in small local facilities does not influence rates.

### Interpretation of findings

With respect to distance as a contributing factor for surgical access, travel times to surgical facilities are associated with CS and hysterectomy rates but not sterilisation. The evident lack of association between distance and sterilisation rates may be explained by the greater availability of this procedure at smaller facilities and surgical outreach camps; the 2015–2016 DHS reports that in rural areas 53% of sterilisation is provided at a level below a government municipal or main private hospital.[14] Surgery in private facilities where capacity is much more dispersed and dominated by non-poor users is less contingent on travel time. This may be linked to the greater number of smaller private facilities that offer surgery so that travel time is less important. Rates of CS are substantially above the 10%–15% level even for the poorest income groups. These higher rates may result partly from women with more complex deliveries seeking care from these facilities but also an excess preference for CS over non-surgical delivery in private facilities, a concern raised in other studies across India.[27]

We find the presence of non-doctor medical staff is generally associated with lower levels of surgical activity.

**Table 3** Costs of delivery care (US$ and % of per capita income)

| | Public | Private | Total | Public | Private | Total |
|---|---|---|---|---|---|---|
| | Cost per delivery | | | % of per capita income | | |
| Non-surgical delivery | | | | | | |
| Facility | $38 | $140 | $50 | 3.3 | 12.0 | 4.3 |
| Transport | $5 | $9 | $6 | 0.4 | 0.8 | 0.5 |
| Total | $44 | $148 | $56 | 3.8 | 12.8 | 4.8 |
| Surgical delivery | | | | | | |
| Facility | $120 | $311 | $199 | 10.3 | 26.8 | 17.1 |
| Transport | $8 | $15 | $11 | 0.7 | 1.3 | 0.9 |
| Total | $128 | $325 | $210 | 11.0 | 28.0 | 18.1 |

state-wise Gross Domestic Product (GDP) obtained from http://mospi.nic.in/data.

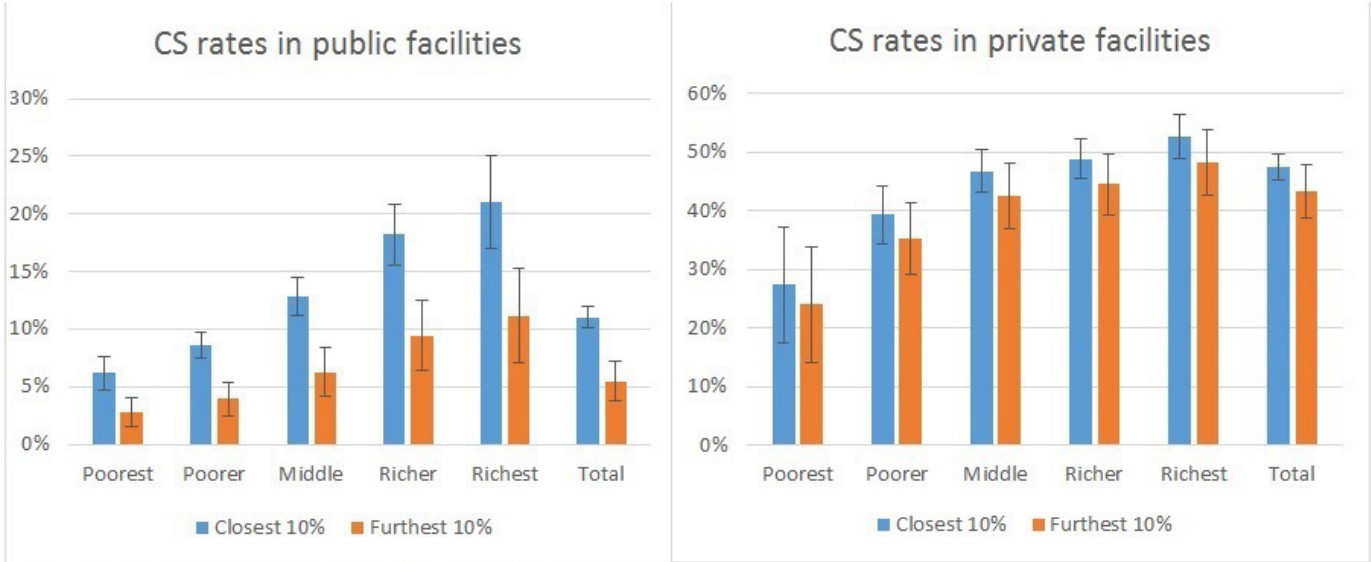

**Figure 5** Caesarean section rate as proportion of all deliveries in public and private facilities. CS, caesarean section (error bars are 95% confidence intervals).

This is supported by other findings: across widely different contexts—for example, the USA and Iran[28 29]—there is evidence that surgical delivery is lower where more care is provided by nurses and midwives. Our study supports this with lower rates of both CS and sterilisation where similar services can be provided by non-surgical alternatives.

The negative association between membership of the RSBY scheme and CS and lack of association with hysterectomy and CS surgery and the state insurance scheme, also found in other studies,[30] remains puzzling given that both form part of the respective insurance packages. One explanation is that since RSBY is largely a scheme for the poor, the negative effect of lower income override any positive impact of holding insurance. Another explanation is that beneficiaries are unware of benefits of insurance. This was suggested in a recent study of RSBY which revealed that lack of awareness of scheme benefits resulted in RSBY beneficiaries accessing paid, private services rather than through RSBY facilities.[31] A further explanation may be that the reimbursement rates for surgery across the states are lower than the costs of services making facilities less likely to accept insured patients. Delays and other issues with the insurance mechanisms also often make private providers reluctant to participate in schemes.[31] The findings suggest that access to surgery is not guaranteed by inclusion in an insurance benefits package. It is also important to ensure that full costs of delivering services are properly compensated particularly for providers that are delivering services to more vulnerable groups.

### Limitations

The study has a number of limitations. The self-reported DHS data do not indicate whether surgery was clinically indicated at an individual level, so drawing definitive conclusions on excessive surgery at an individual level is therefore not possible. Nevertheless, the population estimates suggest rates that for some groups are too high and some too low. The distance variable is a rough estimate of the actual time to a facility because of the geographical displacement in the Geographic Information System (GIS) variable and the assumption that facilities are based in the centre of the district in the absence of precise GIS coordinates for facilities. Finally, the study takes no account of the quality of service provided in facilities, a variable that in itself is likely to be an important but missing determinant of use of services.

### Implications for policy and practice

The huge variation in access to gynaecological and obstetric surgery across North-East India illustrates the persistent importance of factors that both discourage essential surgical care and incentivise some unnecessary surgery. Evidence from a range of contexts suggest that supply-side improvements will not increase use of services without mitigating demand barriers.[32] Cost-effective solutions to alleviate these barriers might either take the form of increasing services in remoter areas (eg, through outreach services plus rural case finding as practised in other contexts),[33] increasing the ability of vulnerable groups to access geographically more accessible private services at low cost or attempting to compensate for the greater travel costs of those living in remoter areas. Elsewhere in South Asia, such compensation has been shown to be important in increasing service use.[34 35] Consideration might be given to incorporating such compensation into financing mechanisms.

While increasing access to services is important for some, for others policy needs to focus on limiting excessive surgery. For CS and hysterectomy, high rates are present for richer and more educated groups but for sterilisation higher use is evident for those with less education. Policy to reduce unnecessary surgery in rich and poor households requires ensuring adequate and accessible staff to

promote and undertake non-surgical alternatives in both rural and urban areas. At the same time, further attention is required to ensure that surgery is accessible when required by mitigating demand-side barriers for those with clear clinical need.

**Acknowledgements** We acknowledge the helpful comments of three independent reviewers. The views expressed in this publication are those of the author(s) and not necessarily those of the NIHR or the UK Department of Health and Social Care.

**Contributors** TE conceived and designed the study undertook the data analysis, contributed to interpretation of the results and was wrote the first draft of the paper. AV helped guide the development of study, interpretation of results and contributed to policy conclusions. She read and edited the final draft. NA provided input on the interpretation of results particularly around clinical aspects of health seeking behaviour. He read and edited the final draft.

**Funding** This research was funded by the National Institute for Health Research (NIHR) (16/137/44) using UK aid from the UK Government to support global health research.

**Competing interests** None declared.

**Patient and public involvement** Patients and/or the public were not involved in the design, or conduct, or reporting, or dissemination plans of this research.

**Patient consent for publication** Not required.

**Ethics approval** The data used in this study are all anonymised and in the public domain and no additional ethics approval was required in order to undertake the study. DHS was downloaded from the Measure website (dhsprogram.com) while district HMIS and census data are available through public facing websites ( censusindia.gov.in; nrhm-mis.nic.in). The funder of the research had no role in the design of the surveys used (NFHS, HMIS or District Census) nor any role in the, analysis and interpretation of data, writing the report or the decision to submit the paper for publication.

**Provenance and peer review** Not commissioned; externally peer reviewed.

**ORCID iDs**
Tim Ensor http://orcid.org/0000-0003-0279-9576
Amrit Virk http://orcid.org/0000-0001-8686-2776
Noel Aruparayil http://orcid.org/0000-0002-2898-772X

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
