## [Reviewer comments · BMJ Open]

ARTICLE DETAILS

TITLE (PROVISIONAL)	Factors influencing use of essential surgical services in North-East India: a cross sectional study of obstetric and gynaecological surgery
AUTHORS	Ensor, Tim; Virk, Amrit; Aruparayil, Noel

VERSION 1 – REVIEW

REVIEWER	Maria Delius LMU Munich University Department Obstetrics and Gynecology, Munich, Germany
REVIEW RETURNED	05-Apr-2020

GENERAL COMMENTS	Thak you for this interesting Research.
---

REVIEWER	Charles Ameh Liverpool School of Tropical Medicine, United Kingdom
REVIEW RETURNED	13-Apr-2020

GENERAL COMMENTS	Thanks for submitting an interesting manuscript that has analysed secondary data to describe availability of essential obstetric/reproductive surgical services in a region of India. There are a few grammatical issues throughout the manuscript. It is unclear in the introduction if this study fills a gap in the literature. In the introduction, the can improve the understanding of readers. The 'study' can not be the subject. Please rephrase...This study aims to determine..... Please review the consistency of the citation style Page 4 line 35, the sentence beginning with Although...appears to be redundant. The methods section is not structured. Adding subheadings or clear paragraphs relate to Study design, setting, description of variable, analysis, ethical considerations etc, is likely to improve readability. Add relevant citation for 'asset-based index or describe what it is. The description of variables needs to be more precise for example density of medical personnel by cadre ie number of medical doctors per 1000 population. How representative are the surveys whose data were included? Please use the appropriate in-text citation for STATA Page 6 line 28...we calculated rather than we calculate Page 6 line 31-Do you mean total time travel to... Page 6 line 38-Please be explicit: No role in the design of NFHS, DHS, HMIS and Census surveys. This way it is different from the study design of your manuscript (which is unclear at present)
---

	Ethics considerations and statement is unclear Page 5 line 50: Please use surgical procedures or operations consistently rather than interchangeably throughout. Page 8 line 11..'The NFHS reported...rather than 'finds... The citation of footnotes and the referencing style conflict. It is unclear if 1, 2, 3 etc in the text refers to the 1, 2, 3 etc in the reference list or the footnote. Results The presentation of results is mixed with the interpretation of results that should be in the discussion section. To improve this section, please consider the use of sub-headings such as Essential reproductive/obstetric surgical rates Factors associated with the availability of reproductive/obstetric surgical procedures 1) supply side. 2) demand-side Discussion I suggest state the key results in the first paragraph, other clear subsections are interpretation of findings, strengthens and limitations, the implication for research and practice, and conclusion. The finding of non-medical staff associated with lower levels of surgical activity is not surprising unless there is any other cadre expected to perform surgery.
--	---

REVIEWER	Chiara Pittalis Royal College of Surgeons in Ireland
REVIEW RETURNED	07-Jul-2020

GENERAL COMMENTS	The study examines a number of demand-side factors affecting obstetric surgical care delivery in NE India. While findings may not be generalisable to other countries, they provide important insights to inform context-specific policy interventions. in general the manuscript is clear and well written, the only suggestions would be:  - to define the term 'substantial surgical facility' for the benefit of the reader - Line 54 pg. 4: explicitly list the individual and household characteristics considered in the study for the benefit of the reader - it may be interesting to have a comparison between elective and emergency procedures if data are available
---

VERSION 1 – AUTHOR RESPONSE

Reviewer 1	
Thank you for this interesting Research	Thank you
Reviewer 2	
There are a few grammatical issues throughout the manuscript.	Thank for noticing this. We have read through the manuscript and corrected some issues.
In the introduction, the can improve the understanding of readers.	Changed to 'aims to examine'

The 'study' can not be the subject. Please rephrase...This study aims to determine.....	
Please review the consistency of the citation style	All are now Vancouver (numbered) style. I have taken out any footnotes.
Page 4 line 35, the sentence beginning with Although...appears to be redundant.	'Although' removed
The methods section is not structured. Adding subheadings or clear paragraphs relate to Study design, setting, description of variable, analysis, ethical considerations etc, is likely to improve readability.	We have added a date, description of variables and ethical considerations sub-titles. Also slightly re ordered the paragraphs.
Add relevant citation for 'asset-based index or describe what it is.	A reference to a World Bank publication has been added.
The description of variables needs to be more precise for example density of medical personnel by cadre ie number of medical doctors per 1000 population.	We have added units e.g. per 1,000 women or population where relevant
How representative are the surveys whose data were included?	There are three sources of data. The NFHS is statistically representative down to the district level. The HMIS and District census are both censuses so should capture all or most information on facilities and other local characteristics. This is clarified in the text.
Please use the appropriate in-text citation for STATA	I have added a reference to: WEBER, S. 2018. A simple command to calculate travel distance and travel time. The State Journal, 17, 962-971.
Page 6 line 28...we calculated rather than we calculate	Changed to 'calculated'
Page 6 line 31-Do you mean total time travel to...	I am not totally sure which phrase this refers to. However I have added 'total' before travel time in the sentence: "This takes pairs of coordinates (longitude and latitude) and accesses the HERE mapping database ^[1] to compute the total travel time using available roads adjusted for speed of traffic."
Page 6 line 38-Please be explicit: No role in the design of NFHS, DHS, HMIS and Census surveys. This way it is different from the study design of your manuscript (which is unclear at present)	I have modified the sentence: " The funder of the research had no role in the of the NFHS, HMIS or District Census nor any role in the study design, collection, analysis and interpretation of data, writing the report or

	the decision to submit the paper for publication.”
Ethics considerations and statement is unclear	Sentence added: “The data used in this study are all anonymised and in the public domain and no additional ethics approval was required in order to undertake the study.”
Page 5 line 50: Please use surgical procedures or operations consistently rather than interchangeably throughout.	4 occurrences of ‘operations’ changed to ‘surgical procedures’
Page 8 line 11..‘The NFHS reported...rather than ‘finds...	Changed to ‘reported’
The citation of footnotes and the referencing style conflict. It is unclear if 1, 2, 3 etc in the text refers to the 1, 2, 3 etc in the reference list or the footnote.	I have converted the footnotes into references so there is now no conflict.
The presentation of results is mixed with the interpretation of results that should be in the discussion section.	I have moved a couple of the interpretations of results in the results section to the discussion section.
Results To improve this section, please consider the use of sub-headings such as Essential reproductive/obstetric surgical rates Factors associated with the availability of reproductive/obstetric surgical procedures 1) supply side. 2) demand-side	I have considered sub-sections but don’t think the section would easily divide like this as the focus is first on the descriptive and then the multivariate results.
Discussion I suggest state the key results in the first paragraph, other clear subsections are interpretation of findings, strengths and limitations, the implication for research and practice, and conclusion.	Key results are presented in the first paragraph. I have added the sub headings as suggested
The finding of non-medical staff associated with lower levels of surgical activity is not surprising unless there is any other cadre expected to perform surgery.	Not surprising perhaps but it does have a policy implication since a greater density of doctors is likely to increase both essential and unnecessary surgery. So we felt it was important to report this finding.
Reviewer 3	
to define the term ‘substantial surgical facility’ for the benefit of the reader	We have taken out the word ‘substantial’ because in fact we look at facilities with varying surgical capacity from 1 of fewer a day up to 2000 or more a year
Line 54 pg. 4: explicitly list the individual and household characteristics considered in the study for the benefit of the reader	A list has been added in the methods (description of variables) section.

it may be interesting to have a comparison between elective and emergency procedures if data are available	Thank you for this good suggestion. Unfortunately this is not possible given the data available
--	---

VERSION 2 – REVIEW

REVIEWER	Chiara Pittalis Royal College of Surgeons in Ireland, Ireland
REVIEW RETURNED	17-Aug-2020

GENERAL COMMENTS	Improved version. You could have elaborated a bit more on the approach statistical analysis in the methods section (e.g. choice of significance levels, etc.) but overall the paper is in order. Applicable international reporting guidelines (such as STROBE) such be applied and reported against.
---

VERSION 2 – AUTHOR RESPONSE

Reviewer	
You could have elaborated a bit more on the approach statistical analysis in the methods section (e.g. choice of significance levels, etc.) but overall the paper is in order.	We have a little more detail on significance, types of variables and commands used to analyse individual associations.
Applicable international reporting guidelines (such as STROBE) such be applied and reported against.	We have gone through the 22 point STROBE checklist and ensured the paper complies with the criteria although the individual criteria are not fully used to structure the sections of the paper.